# Edge based metric dimension of various coffee compounds

**Ali Ahmad**[1], **Ali N. A. Koam**[2], **Muhammad Azeem**[3]*, **Ibtisam Masmali**[1], **Rehab Alharbi**[1]

**1** Department of Information Technology and Security, College of Computer Science and Information Technology, Jazan University, Jazan, Saudi Arabia, **2** Department of Mathematics, College of Science, Jazan University, New Campus, Jazan, Saudi Arabia, **3** Department of Mathematics, Riphah International University, Lahore, Pakistan

* azeemali7009@gmail.com

## Abstract

An important dietary source of physiologically active compounds, coffee also contains phenolic acids, diterpenes, and caffeine. According to a certain study, some coffee secondary metabolites may advantageously modify a number of anti-cancer defense systems. This research looked at a few coffee chemical structures in terms of edge locating numbers or edge metric size to better understand the mechanics of coffee molecules. Additionally, this research includes graph theoretical properties of coffee chemical structures. The chemicals found in coffee, such as caffeine, diterpene or cafestol, kahweol, chlorogenic, caffeic, gallo-tannins, and ellagitannins, are especially examined in these publications.

## 1 Introduction

Due to its stimulating properties, aroma, and taste, coffee is one of the most favored beverages in the world. However, the general public does not have a particularly good opinion of coffee as a food that may promote health [1]. Although under certain circumstances, such as maternity on intolerance to certain coffee compounds, coffee can have a number of advantageous health impacts [2–5]. It is fascinating to note that a new thinking study found that coffee drinking may alter 297 genes' development in healthy women in various aspects [6], therefore impacting immunological and nutritional processes. Numerous pieces of data suggest that drinking coffee lowers the risk of cardiac illnesses [4, 7], diabetes [8], neurodegenerative disease [9, 10], cancer [3, 11], also to determine in cancer rates and overall morality [4, 12, 13]. Consistent coffee drinking protects DNA integrity by reducing unscheduled DNA strand breaks on a molecular level [14].

Biologically active substances, many of which have anti-cancer potential, may be found in coffee. Epidemiological studies on living beings, which mostly examine caffeine and coffee use (caffeinated or decaffeinated), present somewhat contradictory findings for breast, uterine, and ovarian cancer. In addition to caffeine, phenolic acids, and diterpenes, roasted coffee beans contain a variety of compounds from other chemical families. Their presence in a coffee brew mostly depends on the kind of bean used (Coffea arabica vs. Coffea canephora var. robusta), roasting conditions, and brewing techniques. Diterpenes like cafestol and kahweol,

**Data Availability Statement:** This paper does not report data and the data availability policy is not applicable.

**Funding:** The authors extend their appreciation to the Deputyship for Research and Innovation,

Ministry of Education in Saudi Arabia for funding this research work through the project number ISP22-6. The funders had no role in study design, data collection and analysis, decision to publish, or preparation of the manuscript.

**Competing interests:** The authors have declared that no competing interests exist.

which are unique to coffee, are found in coffee brews [15, 16]. Tea, cola-style drinks, and chocolate all include other substances, such as caffeine.

A new parameter related to metric dimension introduced in [17], few generalized results of metric dimension in terms of diameter regarding the bipartite type of graphs found in [18]. Extremal results in terms of bounded metric dimension are discussed in [19]. A weighted metric dimension is introduced in [20], they discussed a few hard and easy cases to tackle and also linked this to the basic metric dimension. Few lower bounds on the metric dimension are found in [21]. Fuzzy graph theory is found in [22], in which they discussed rough graph's metric dimension. Graph's neighborhood metric dimensions are discussed in [23]. It is also another new resolvability parameter like the above. In [24], the patched network is discussed in terms of metrics and other related parameters. They proved that this network has a parametric metric dimension. Breast cancer chemical structures are discussed in [25] and in [26] also discussed partition dimension, in this they discussed some generalized classes of graphs.

In [24], the patched network is discussed in terms of metrics and other related parameters. They proved that this network has a parametric edge metric dimension. Polycyclic aromatic hydrocarbons are studied in [27], and they proved that metric, edge metric and fault-tolerant versions are constant and do not depend on the number of vertices of this chosen structure. Few graphs having metric dimensions greater than edge metric dimension are studied in [28]. Particularly Mobius and in general ladder related networks are discussed in [29]. They proved that edge metric dimension of all the chosen structures is constant. Edge and its fault-tolerant version of metric dimension are studied in [30], they studied these parameters for the hollow coronoid structures. Similarly, all the resolvability parameters for the benzenoid tripod structure are found in [31]. Metric resolvability and its fault-tolerant version for the Quartz structure found in [32]. An approximated version of the algorithm for computing the edge metric dimension is found in [33]. In [34], the planar graph's edge metric dimension is discussed, they proved that this structure has finite edge resolvability and does not depend on the number of vertices.

The idea of the locating number/metric dimension was first introduced by Slater [35], followed by Harary and Melter [36]. Since then, there has been a great deal of research done on the locating number problem. Following then, this concept was adopted and renamed in numerous additional ways by numerous researchers. The idea of locating set is renamed as the metric dimension in the study work of, [37]. While scholars called the same notion with metric basis or resolving set in purely theoretical terms. A more modern definition of locating set was developed in the last ten years. Kelenc et al. [38] asked the question about the relation among resolvability parameters of a graph. Motivated by the question in this paper, we study the metric dimension of a few coffee structures, namely called Caffeine, diterpene or cafestol and kahweol, Chlorogenic, Caffeic, Gallotannins, and Ellagitannins.

Metric dimensions are used in various hypothetical problems or at least linked, like pharmaceutical chemistry, image processing, decoding complicated games, robot roving, and combinatorial optimization. All these can be found in [35, 39, 40]. Moreover, metric dimension applications were found for the polymer industry in [41]. Similarly, these parameters beneficial for electronic devices are found in [42]. For more recent work on this topic, we refer to see [43–48].

Atoms in drug architecture are referred to as vertices and the bonds that connect them as edges. While $V$ and $E$ in the chemical graph are referred to as the vertex and the edge set, respectively, graph $G(V, E)$ is assumed to be linked, finite and simple. The number of vertices next to $u \in V(G)$, represented by $d_u$, is the degree of a graph vertex. The degree of a vertex in a graph and the valence of a chemical are notions that are intrinsically connected in chemistry [49]. The methodology used in this work is summarized in the flowchart shown in Fig 1. Moreover, the following definitions are much needed to understand the concept used in this work.

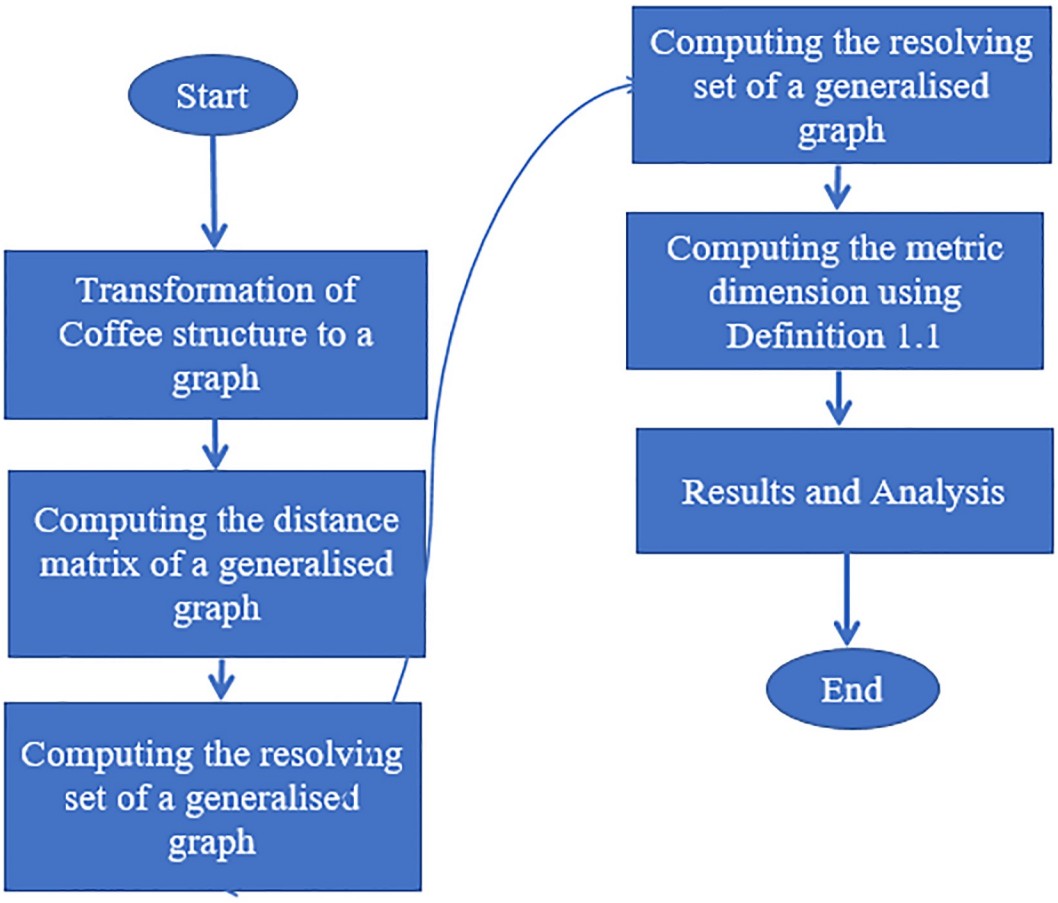

**Fig 1. Flowchart to compute the metric dimension of generalised graph.**

**Definition 0.1**. [30] "*Suppose $G(V(G), E(G))$ is an undirected graph of a chemical network with $V(G)$ is called as set of vertex set and $E(G)$ is the edge set. The distance between two vertices $v_1, v_2 \in V(G)$, denoted as $d(v_1, v_2)$ is the minimum count of edges between $v_1 - v_2$ path.*"

**Definition 0.2**. [30] "*Suppose $R \subset V(G)$ is the subset of vertex set and defined as $R = \{v_1, v_2, \ldots, v_s\}$, and let a vertex $v \in V(G)$. The representations $r(v|R)$ of a vertex $v$ with respect to $R$ is actually a $s$–ordered distances $(d(v, v_1), d(v, v_2), \ldots, d(v, v_s))$. If each vertex from $V(G)$ have unique representations according to the ordered subset $R$, then this subset renamed as a resolving set of network G. The minimum numbers of the elements in the subset R is actually the metric dimension (locating number) of G and it is denoted by the term $dim(G)$.*"

**Definition 0.3**. [30] "*A principal node $v \in V(G)$ and an edge $e = v_1 v_2 \in E(G)$, the distance between $v$ and $e$ is defined as $d(e, v) = \min\{d(v_1, v), d(v_2, v)\}$. Suppose $R_e \subset V(G)$ is the subset of vertex set and defined as $R_e = \{v_1, v_2, \ldots, v_s\}$, and an edge $e \in E(G)$. The representations $r(e|R_e)$ of an edge $e$ with respect to $R_e$ is actually a $s$–tuple distances $(d(e, v_1), d(e, v_2), \ldots, d(e, v_s))$. If each edge from $E(G)$ have unique representations according to $R_e$, then $R_e$ is called an edge metric resolving set of network G. The minimum count of the elements in $R_e$ is called the edge metric dimension (edge locating number) of G and it is represented by $dim_e(G)$.*"

The next section considered the main results or computational work of this article. Next to this section, we added the impacts of this study, and the conclusion is drawn. In the end, sufficient references are added for the reader of this study.

## 2 Main results

Caffeine is a methylxanthine alkaloid that is heat-stable and is most commonly found in coffee, tea, guarana, cola-style soft beverages, cacao, and chocolate. It readily dissolves in hot water. Coffee may contain 19–803 mg of caffeine per serving, depending on the technique of brewing (boiling, filtering, French pressing) and the kind of coffee preparation (ground, instant). Conversely, decaffeinated coffee has a significantly lower caffeine content-9 mg/serving. By using the methodology presented in the flowchart shown in Fig 1, we have developed the graph of Caffeine structure (both are shown in Figs 2 and 3). The developed graph's order and size are finite and counted as $|V(G_{\text{Caffeine}})| = 14$ and $|E(G_{\text{Caffeine}})| = 15$. Moreover, the vertices and edges labeling used in the main results are shown by the following equations.

$$V(G_{\text{Caffeine}}) = \{v_\sigma : \sigma = 1, 2, \ldots, 14\},$$
$$E(G_{\text{Caffeine}}) = \{v_\sigma v_{\sigma+1} : \sigma = 1, 2, \ldots, 8\} \cup \{v_1 v_9, \ v_3 v_7, v_4 v_{12}, \ v_2 v_{11}, \ v_1 v_{10}, \ v_9 v_{14}, \ v_8 v_{13}\}.$$

**Theorem 1.1**. *Let* $G_{Caffeine}$ *be a graph of caffeine structure. Then the edge metric dimension of* $G_{Caffeine}$ *is two.*

*Proof.* To prove the edge metric dimension two for the caffeine structure, by using the methodology presented in Fig 1, let an edge resolving set $R_e(G_{\text{Caffeine}}) = \{v_{10}, v_{12}\}$. Now, to fulfill the definition of unique code, we have presented the representations of each edge and proved that each edge has unique code with the help of a chosen edge resolving set $R_e$. For the clearer view of representations the equation is developed as: $r(e_\sigma|R_e) = (d(i_\sigma, v_{10}), d(e_\sigma, v_{12}))$,

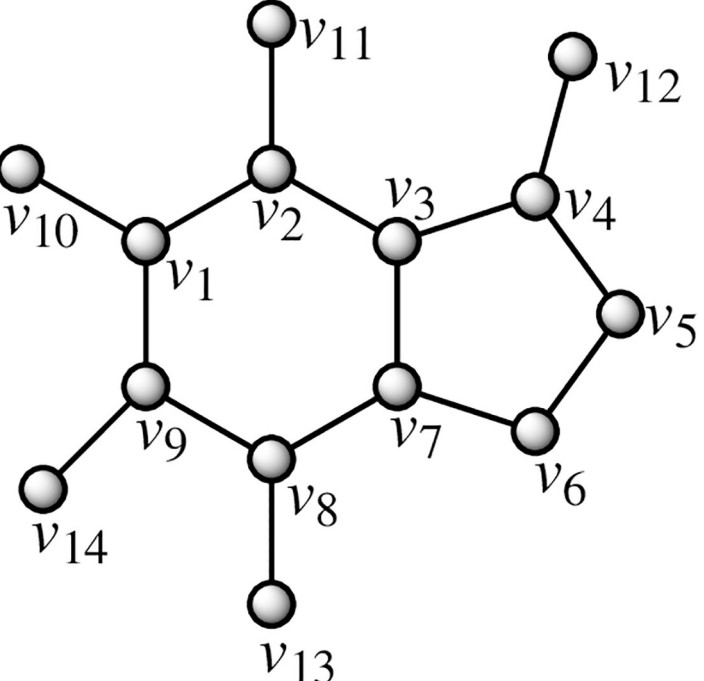

**Fig 2. Caffeine corresponding graph.**

**Fig 3. Caffeine structure.**

and also given in the Table 1.

$$r(v_\gamma v_{\gamma+1}|R_e) = \begin{cases} (\gamma, \ 4 - \gamma) & \text{when } \gamma = 1, 2, 3 \\ (\gamma, \ \gamma - 3) & \text{when } \gamma = 4, 5 \\ (10 - \gamma, \ \gamma - 3) & \text{when } \gamma = 6 \\ (10 - \gamma, \ \gamma - 4) & \text{when } \gamma = 7, 8. \end{cases}$$

By using the definition, we have received the representations ($r(e_\sigma|R_e)$) of all the edges of the caffeine structure, and we can see that there are no two edges having the same identifications or $r(e_\sigma|R_e)$. So, we can conclude that caffeine structure has two edge metric dimensions.

Four isoprene units make up the chemical molecule known as a diterpene. They are mostly represented in coffee by the lipid fraction-specific compounds cafestol and kahweol. This lipid component is absorbed by the cellulose paper filter in coffee filters [15]. The amount of cafestol

**Table 1. Representations of the edges given in the Fig 2.**

| $e$ | $r(e|R_e)$ | $r(e|R_e)$ |
|---|---|---|
| $v_1 v_9$ | 1 | 4 |
| $v_3 v_7$ | 3 | 2 |
| $v_4 v_{12}$ | 4 | 0 |
| $v_2 v_{11}$ | 2 | 3 |
| $v_1 v_{10}$ | 0 | 4 |
| $v_9 v_{14}$ | 2 | 5 |
| $v_8 v_{13}$ | 3 | 4 |

and kahweol in filtered coffee is therefore quite little. Coffea arabica has been discovered to have more diterpenes than C. robusta [50]. Depending on how the coffee is prepared, there are between 0.02 and 9.2 mg of cafestol and 0.02-7.2 mg of kahweol in each cup. By using the methodology presented in the flowchart shown in Fig 1, we have developed the graph of cafestol or kahweol structure (both are shown in Figs 4 and 5). The developed graph's order and size are finite and counted as $|V(G_{\text{Cafestol}})| = 23$ and $|E(G_{\text{Cafestol}})| = 27$. Moreover, the vertices

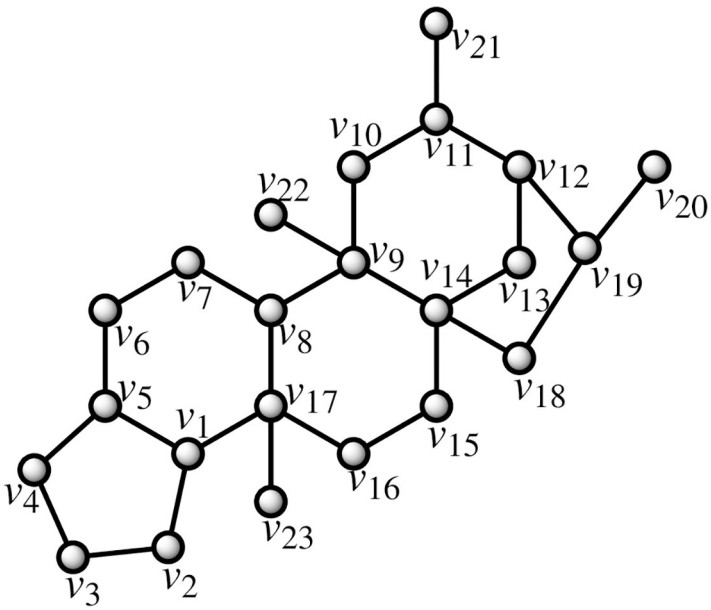

**Fig 4. Cafestol corresponding graph.**

**Fig 5. Cafestol structure.**

and edges labeling used in the main results are shown by the following equations.

$$V(G_{\text{Cafestol}}) \quad = \{v_\sigma \ : \ \sigma = 1, 2, \ldots, 23\},$$

$$E(G_{\text{Cafestol}}) \quad = \{v_\sigma v_{\sigma+1} \ : \ \sigma = 1, 2, \ldots, 16, 18, 19\} \cup \{v_1 v_5, \ v_8 v_{17}, v_{17} v_{23}, \ v_9 v_{14}, \ v_{14} v_{18}, \ v_{12} v_{19}, \ v_{11} v_{21},$$

$$v_9 v_{22}, \ v_1 v_{17}\}.$$

**Theorem 1.2**. *Let $G_{Cafestol}$ be a graph of cafestol structure. Then the edge locating number of $G_{Cafestol}$ is three.*

*Proof*. To prove the edge metric dimension three for the cafestol structure, by using the methodology presented in the Fig 1, let an edge resolving set $R_e(G_{\text{Cafestol}}) = \{v_3, v_{21}, v_{23}\}$. Now, to fulfil the definition of unique code, we have presented the representations of each edge and proved that each edge has unique code with the help of a chosen edge resolving set $R_e$. For the clearer view of representations the equation is developed as: $r(e_\sigma | R_e) = (d(e_\sigma, v_3), d(e_\sigma, v_{21}), d(e_\sigma, v_{23}))$, and also given in the Table 2.

$$r(v_\gamma v_{\gamma+1} | R_e) = \begin{cases} (2 - \gamma, \ 5 + \gamma, \ \gamma + 1) & \text{when } \gamma = 1, 2 \\ (\gamma - 3, \ 11 - \gamma, \ 7 - \gamma) & \text{when } \gamma = 3, 4 \\ (\gamma - 3, \ 11 - \gamma, \ \gamma - 2) & \text{when } \gamma = 5 \\ (\gamma - 3, \ 11 - \gamma, \ 9 - \gamma) & \text{when } \gamma = 6, 7 \\ (\gamma - 4, \ 11 - \gamma, \ \gamma - 6) & \text{when } \gamma = 8, 9, 10 \\ (\gamma - 4, \ \gamma - 10, \ \gamma - 6) & \text{when } \gamma = 11 \\ (19 - \gamma, \ \gamma - 10, \ 17 - \gamma) & \text{when } \gamma = 12, 13, 14, 15 \\ (19 - \gamma, \ 21 - \gamma, \ 17 - \gamma) & \text{when } \gamma = 16 \\ (\gamma - 11, \ 3, \ \gamma - 13) & \text{when } \gamma = 18, 19. \end{cases}$$

By using the definition, we have received the representations ($r(e_\sigma|R_e)$) of all the edges of cafestol structure, and we can see that there are no two edges having the same identifications or $r(e_\sigma|R)$. So, we can conclude that cafestol structure has three edge metric dimension.

Chlorogenic, and ferulic acids are phenolic chemicals classified as hydroxycinnamic acids according to their chemical structure. Quinic acids can be esterified to form chlorogenic acid

**Table 2. Representations of the edges given in the Fig 4.**

| $e$ | $r(e|R_e)$ | $r(e|R_e)$ | $r(e|R_e)$ |
|---|---|---|---|
| $v_1 v_5$ | 2 | 6 | 2 |
| $v_8 v_{17}$ | 3 | 4 | 1 |
| $v_{17} v_{23}$ | 3 | 5 | 0 |
| $v_9 v_{14}$ | 5 | 3 | 3 |
| $v_{14} v_{18}$ | 6 | 4 | 4 |
| $v_{12} v_{19}$ | 8 | 2 | 6 |
| $v_{11} v_{21}$ | 7 | 0 | 5 |
| $v_9 v_{22}$ | 5 | 3 | 3 |
| $v_1 v_{17}$ | 2 | 5 | 1 |

(3-caffeoylquinic acid). There are a wide variety of hydroxycinnamic acids, which are also typical of green coffee, in nature. They are lignin biosynthesis' intermediaries. Although hydroxycinnamic acids are present in almost all plant-based diets, significant levels have also been discovered in coffee. Therefore, in nations with high coffee consumption, coffee may be the primary source of hydroxycinnamic acids [51]. By using the methodology presented in the flowchart shown in the Fig 1, we have developed the graph of Chlorogenic structure (both are shown in Figs 6 and 7). The developed graph's order and size is finite and counted as $|V(G_{\text{Chlorogenic}})| = 25$ and $|E(G_{\text{Chlorogenic}})| = 26$. Moreover, the vertices and edges labelling used in the main results are shown by the following equations.

$$V(G_{\text{Chlorogenic}}) = \{v_\sigma \ : \ \sigma = 1, 2, \ldots, 25\},$$
$$E(G_{\text{Chlorogenic}}) = \{v_\sigma v_{\sigma+1} \ : \ \sigma = 1, 2, \ldots, 15, 21, 22\} \cup \{v_1 v_6, \ v_1 v_{19}, v_2 v_{20}, \ v_4 v_{22}, \ v_4 v_{24}, \ v_8 v_{25}, \ v_{11} v_{16},$$
$$v_{14} v_{17}, \ v_{13} v_{18}\}.$$

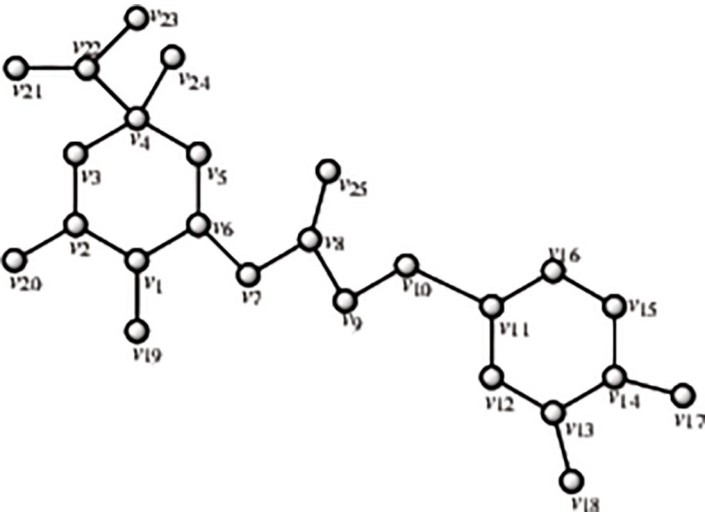

**Fig 6. Chlorogenic corresponding graph.**

**Fig 7. Chlorogenic structure.**

**Theorem 1.3**. *Let $G_{Chlorogenic}$ be a graph of chlorogenic structure. Then the edge locating number of $G_{Chlorogenic}$ is three.*

*Proof.* To prove the edge metric dimension three for the chlorogenic structure, by using the methodology presented in the Fig 1, let an edge resolving set $R_e(G_{Chlorogenic}) = \{v_{18}, v_{19}, v_{21}\}$. Now, to fulfil the definition of unique code, we have presented the representations of each edge and proved that each edge has unique code with the help of a chosen edge resolving set $R_e$. For the clearer view of representations the equation is developed as: $r(e_\sigma|R_e) = (d(e_\sigma, v_{18}), d(e_\sigma, v_{19}), d(e_\sigma, v_{21}))$, and also given in the Table 3.

$$r(v_\gamma v_{\gamma+1}|R_e) = \begin{cases} (8+\gamma, \ \gamma, \ 5-\gamma) & \text{when } \gamma = 1,2 \\ (13-\gamma, \ \gamma, \ 5-\gamma) & \text{when } \gamma = 3 \\ (13-\gamma, \ 7-\gamma, \ \gamma-2) & \text{when } \gamma = 4,5 \\ (13-\gamma, \ \gamma-4, \ \gamma-2) & \text{when } \gamma = 6,7,\ldots,12 \\ (\gamma-12, \ \gamma-4, \ \gamma-2) & \text{when } \gamma = 13 \\ (\gamma-12, \ 23-\gamma, \ 25-\gamma) & \text{when } \gamma = 14,15 \\ (11, \ 5, \ \gamma-21) & \text{when } \gamma = 21,22. \end{cases}$$

By using the definition, we have received the representations ($r(e_\sigma|R)$) of all the edges of chlorogenic structure, and we can see that there are no two edges having the same identifications or $r(e_\sigma|R)$. So, we can conclude that chlorogenic structure has three edge metric dimension.

Caffeic, and ferulic acids are phenolic chemicals classified as hydroxycinnamic acids according to their chemical structure. Caffeic acid (3,4-dihydroxycinnamic acid) is converted in plants into ferulic acid (4-hydroxy-3-methoxy cinnamic acid). Caffeic and quinic acids can be esterified to form chlorogenic acid (3-caffeoylquinic acid). There are a wide variety of hydroxycinnamic acids, which are also typical of green coffee, in nature. It's worth noting that caffeic acid has lately become linked to increased ovarian cancer cell resilience to therapy [52]. By using the methodology presented in the flowchart shown in the Fig 1, we have developed the graph of Caffeic structure (both are shown in the Figs 8 and 9). The developed graph's order and size is finite and counted as $|V(G_{Caffeic})| = 13$ and $|E(G_{Caffeic})| = 13$. Moreover, the vertices and edges labelling used in the main results are shown by the following equations.

$$V(G_{Caffeic}) = \{v_\sigma \ : \ \sigma = 1,2,\ldots,13\},$$
$$E(G_{Caffeic}) = \{v_\sigma v_{\sigma+1} \ : \ \sigma = 1,2,\ldots,9\} \cup \{v_1 v_6, \ v_9 v_{11}, v_2 v_{13}, \ v_3 v_{12}\}.$$

**Table 3. Representations of the edges given in the Fig 6.**

| $e$ | $r(e|R_e)$ | $r(e|R_e)$ | $r(e|R_e)$ |
|---|---|---|---|
| $v_1 v_6$ | 8 | 1 | 4 |
| $v_1 v_{19}$ | 9 | 0 | 5 |
| $v_2 v_{20}$ | 10 | 2 | 4 |
| $v_4 v_{22}$ | 10 | 4 | 2 |
| $v_4 v_{24}$ | 10 | 4 | 2 |
| $v_8 v_{25}$ | 6 | 4 | 6 |
| $v_{11} v_{16}$ | 3 | 7 | 9 |
| $v_{14} v_{17}$ | 2 | 10 | 12 |
| $v_{13} v_{18}$ | 0 | 9 | 11 |

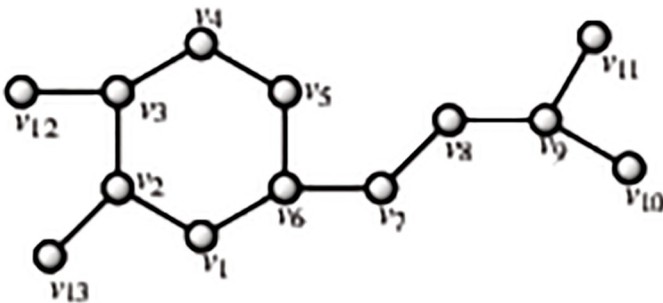

**Fig 8. Caffeic corresponding graph.**

OH

O

OH

O

CH₃

*Ferulic acid*

**Fig 9. Caffeic structure.**

**Theorem 1.4**. *Let $G_{Caffeic}$ be a graph of caffeic structure. Then the edge locating number of $G_{Caffeic}$ is two.*

*Proof.* To prove the edge metric dimension two for the caffeic structure, by using the methodology presented in the Fig 1, let an edge resolving set $R_e(G_{Caffeic}) = \{v_{11}, v_{13}\}$. Now, to fulfil the definition of unique code, we have presented the representations of each edge and proved that each edge has unique code with the help of a chosen edge resolving set $R_e$. For the clearer view of representations the equation is developed as: $r(e_\sigma|R_e) = (d(e_\sigma, v_{11}), d(e_\sigma, v_{13}))$, and also given in the Table 4.

$$r(v_\gamma v_{\gamma+1}|R_e) = \begin{cases} (\gamma + 1, \ \gamma) & \text{when } \gamma = 1 \\ (\gamma + 1, \ \gamma - 1) & \text{when } \gamma = 2 \\ (9 - \gamma, \ \gamma - 1) & \text{when } \gamma = 3, 4 \\ (9 - \gamma, \ \gamma - 2) & \text{when } \gamma = 5 \\ (9 - \gamma, \ \gamma - 3) & \text{when } \gamma = 6, 7, 8. \end{cases}$$

By using the definition, we have received the representations ($r(e_\sigma|R)$) of all the edges of caffeic structure, and we can see that there are no two edges having the same identifications or $r(e_\sigma|R)$. So, we can conclude that caffeic structure has two edge metric dimension.

**Table 4. Representations of the edges given in the Fig 8.**

| $e$ | $r(e\|R_e)$ | $r(e\|R_e)$ |
|---|---|---|
| $v_1 v_6$ | 4 | 2 |
| $v_2 v_{13}$ | 6 | 0 |
| $v_3 v_{12}$ | 7 | 2 |
| $v_9 v_{11}$ | 0 | 6 |

Gallotannins are organic polymers that are created naturally when the hydroxyl groups of D-glucose and gallic acid are esterified and then connected together in polymer network by what are known as "depside" linkages. The word "depside" refers to polyphenols, which can be any chemical, that are made up of two or more monoaromatic units connected by an ester bond. Shikimic acid and quinic acid are esterified with gallic acid to produce different kinds of gallotannins. These gallotannins have a single free carboxylic group in the quinic moiety and, like those made from glucose, a variety of depside bonds connecting their galloyl residues, giving them a highly diverse structural makeup [53, 54]. By using the methodology presented in the flowchart shown in the Fig 1, we have developed the graph of Gallotannins structure (both are shown in Figs 10 and 11). The developed graph's order and size is finite and counted as $|V(G_{\text{Gallotannins}})| = 23$ and $|E(G_{\text{Gallotannins}})| = 24$. Moreover, the vertices and edges labelling used in the main results are shown by the following equations.

$$V(G_{\text{Gallotannins}}) = \{v_\sigma \ : \ \sigma = 1, 2, \ldots, 23\},$$
$$E(G_{\text{Gallotannins}}) = \{v_\sigma v_{\sigma+1} \ : \ \sigma = 1, 2, \ldots, 14, 18\} \cup \{v_{13}v_{16}, \ v_9 v_{14}, v_{12}v_{17}, \ v_{10}v_{18}, \ v_7 v_{20}, \ v_1 v_6, \ v_2 v_{23},$$
$$v_3 v_{22}, \ v_4 v_{21}\}.$$

**Theorem 1.5**. *Let $G_{\text{Gallotannins}}$ be a graph of Gallotannins structure. Then the edge locating number of $G_{\text{Gallotannins}}$ is three.*

*Proof.* To prove the edge metric dimension three for the Gallotannins structure, by using the methodology presented in the Fig 1, let an edge resolving set $R_e(G_{\text{Gallotannins}}) = \{v_{15}, v_{19}, v_{23}\}$. Now, to fulfil the definition of unique code, we have presented the representations of each edge and proved that each edge has unique code with the help of a chosen edge resolving set $R_e$. For the clearer view of representations the equation is developed as: $r(e_\sigma|R) = (d(e_\sigma, v_{15}), d(e_\sigma, v_{19}), d(e_\sigma, v_{23}))$, and also given in the Table 5.

$$r(v_\gamma v_{\gamma+1}|R_e) = \begin{cases} (5+\gamma, \ 6+\gamma, \ \gamma) & \text{when } \gamma = 1 \\ (5+\gamma, \ 6+\gamma, \ \gamma-1) & \text{when } \gamma = 2 \\ (10-\gamma, \ 11-\gamma, \ \gamma-1) & \text{when } \gamma = 3, 4 \\ (10-\gamma, \ 11-\gamma, \ \gamma-2) & \text{when } \gamma = 5 \\ (10-\gamma, \ 11-\gamma, \ \gamma-3) & \text{when } \gamma = 6, 7, 8 \\ (\gamma-7, \ 11-\gamma, \ \gamma-3) & \text{when } \gamma = 9, 10 \\ (\gamma-14, \ \gamma-8, \ \gamma-3) & \text{when } \gamma = 11 \\ (\gamma-14, \ \gamma-8, \ 20-\gamma) & \text{when } \gamma = 12 \\ (\gamma-14, \ 4, \ 20-\gamma) & \text{when } \gamma = 13 \\ (\gamma-14, \ 4, \ 8) & \text{when } \gamma = 14 \\ (4, \ 0, \ 8) & \text{when } \gamma = 18. \end{cases}$$

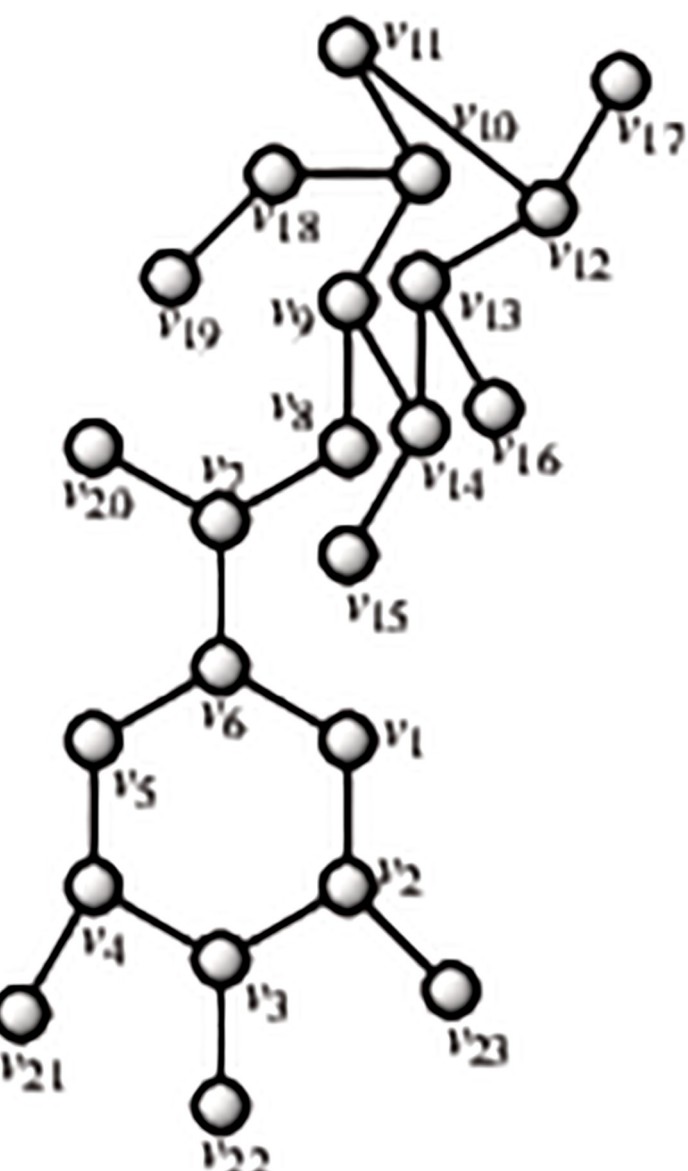

**Fig 10. Gallotannins corresponding graph.**

By using the definition, we have received the representations ($r(e_\sigma|R_e)$) of all the edges of Gallotannins structure, and we can see that there are no two edges having the same identifications or $r(e_\sigma|R_e)$. So, we can conclude that Gallotannins structure has three edge metric dimension.

Ellagitannins (ETs) are esters of polyols like glucose or quinic acid and hexahydroxydiphenic acid. The freed ET molecule from hexahydroxydiphenic acid undergoes hydrolysis and then spontaneously rearranges into the water-insoluble substance ellagic acid. Numerous derivatives of ETs are produced through methylation, glycosylation, and methoxylation in plants [54]. By using the methodology presented in the flowchart shown in the Fig 1, we have developed the graph of Ellagitannins structure (both are shown in Figs 12 and 13). The developed graph's order and size is finite and counted as $|V(G_{\text{Ellagitannins}})| = 34$ and $|E(G_{\text{Ellagitannins}})| = 37$. Moreover, the vertices and edges labelling used in the main results are

# Gallotannins

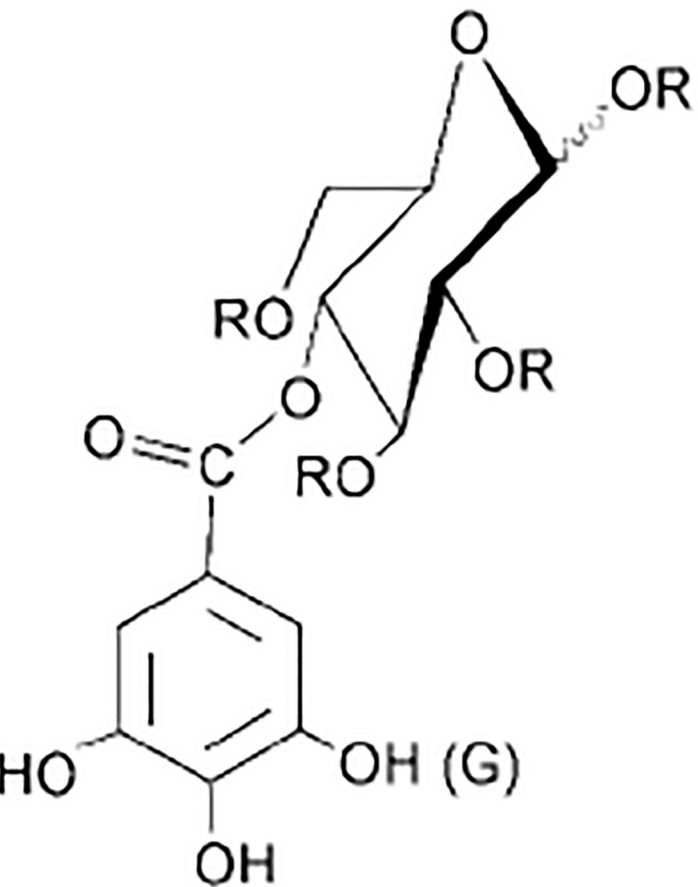

**Fig 11. Gallotannins structure.**

**Table 5. Representations of the edges given in the Fig 10.**

| $e$ | $r(e\|R_e)$ | $r(e\|R_e)$ | $r(e\|R_e)$ |
|---|---|---|---|
| $v_{13}v_{16}$ | 2 | 5 | 8 |
| $v_9v_{14}$ | 1 | 3 | 6 |
| $v_2v_{23}$ | 7 | 8 | 0 |
| $v_{12}v_{17}$ | 3 | 4 | 9 |
| $v_4v_{21}$ | 7 | 8 | 3 |
| $v_{10}v_{18}$ | 3 | 1 | 7 |
| $v_7v_{20}$ | 4 | 5 | 4 |
| $v_1v_6$ | 5 | 6 | 2 |

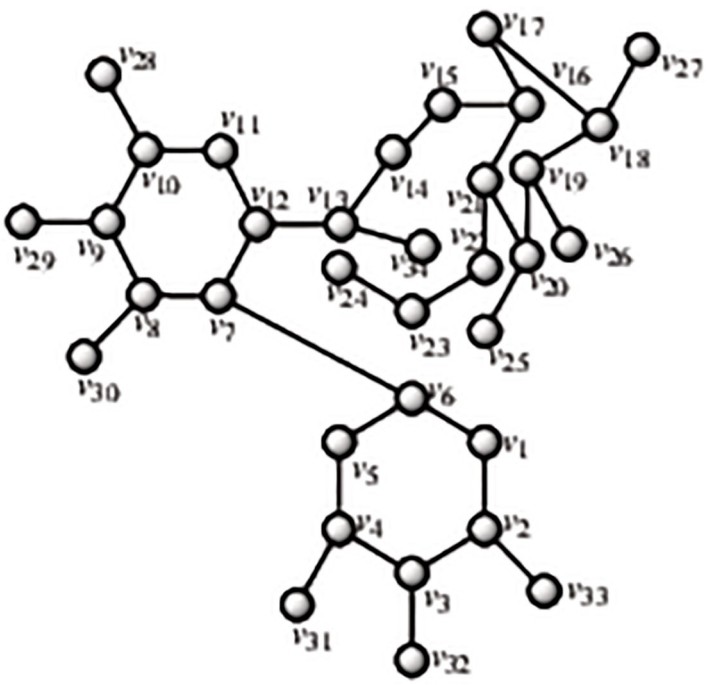

**Fig 12. Ellagitannins corresponding graph.**

# Ellagitannins

**Fig 13. Ellagitannins structure.**

shown by the following equations.

$$
\begin{aligned}
V(G_{\text{Ellagitannins}}) \quad &= \{v_\sigma \ : \ \sigma = 1, 2, \ldots, 34\}, \\
E(G_{\text{Ellagitannins}}) \quad &= \{v_\sigma v_{\sigma+1} \ : \ \sigma = 1, 2, \ldots, 23\} \cup \{v_1 v_6, \ v_2 v_{33}, v_3 v_{32}, \ v_4 v_{31}, \ v_7 v_{12}, \ v_8 v_{30}, \ v_9 v_{29}, \ v_{10} v_{28}, \\
& \quad v_{13} v_{34}, \ v_{16} v_{21}, \ v_{18} v_{27}, \ v_{19} v_{26}, \ v_{20} v_{25}, \ v_{16} v_{21}\}.
\end{aligned}
$$

**Theorem 1.6**. *Let $G_{Ellagitannins}$ be a graph of ellagitannins structure. Then the edge locating number of $G_{Ellagitannins}$ is three.*

*Proof.* To prove the edge metric dimension three for the ellagitannins structure, by using the methodology presented in the Fig 1, let an edge resolving set $R_e(G_{\text{Ellagitannins}}) = \{v_{27}, v_{29}, v_{33}\}$. Now, to fulfil the definition of unique code, we have presented the representations of each edge and proved that each edge has unique code with the help of a chosen edge resolving set $R_e$. For the clearer view of representations the equation is developed as: $r(e_\sigma|R_e) = (d(e_\sigma, v_{27}), d(e_\sigma, v_{29}), d(e_\sigma, v_{33}))$, and also given in the Table 6.

$$
r(v_\gamma v_{\gamma+1}|R_e) = \begin{cases}
(9 + \gamma, \ 4 + \gamma, \ \gamma) & \text{when } \gamma = 1 \\
(9 + \gamma, \ 4 + \gamma, \ \gamma - 1) & \text{when } \gamma = 2 \\
(14 - \gamma, \ 9 - \gamma, \ \gamma - 1) & \text{when } \gamma = 3, 4 \\
(14 - \gamma, \ 9 - \gamma, \ \gamma - 2) & \text{when } \gamma = 5, 6 \\
(1 + \gamma, \ 9 - \gamma, \ \gamma - 2) & \text{when } \gamma = 7, 8 \\
(18 - \gamma, \ \gamma - 8, \ 6) & \text{when } \gamma = 9 \\
(18 - \gamma, \ \gamma - 8, \ 16 - \gamma) & \text{when } \gamma = 10, 11 \\
(18 - \gamma, \ \gamma - 8, \ \gamma - 7) & \text{when } \gamma = 12, 13, \ldots, 17 \\
(\gamma - 17, \ \gamma -, \ \gamma - 7) & \text{when } \gamma = 18 \\
(\gamma - 17, \ 29 - \gamma, \ 30 - \gamma) & \text{when } \gamma = 19, 20 \\
(\gamma - 17, \ \gamma - 12, \ \gamma - 11) & \text{when } \gamma = 21, 22, 23.
\end{cases}
$$

By using the definition, we have received the representations ($r(e_\sigma|R_e)$) of all the edges of ellagitannins structure, and we can see that there are no two edges having the same

**Table 6. Representations of the edges given in the Fig 12.**

| $e$ | $r(e|R_e)$ | $r(e|R_e)$ | $r(e|R_e)$ |
|---|---|---|---|
| $v_1 v_6$ | 9 | 9 | 2 |
| $v_2 v_{33}$ | 11 | 6 | 0 |
| $v_3 v_{32}$ | 12 | 7 | 2 |
| $v_4 v_{31}$ | 11 | 6 | 3 |
| $v_7 v_{12}$ | 7 | 3 | 4 |
| $v_9 v_{29}$ | 8 | 2 | 5 |
| $v_{10} v_{28}$ | 9 | 0 | 6 |
| $v_{13} v_{34}$ | 9 | 2 | 7 |
| $v_{16} v_{21}$ | 6 | 5 | 6 |
| $v_{18} v_{27}$ | 3 | 8 | 9 |
| $v_{19} v_{26}$ | 0 | 10 | 11 |
| $v_{20} v_{25}$ | 2 | 10 | 12 |
| $v_{20} v_{25}$ | 3 | 10 | 11 |

identifications or $r(e_\sigma|R_e)$. So, we can conclude that ellagitannins structure has three edge metric dimension.

## 3 Impact of the study of the edge-based metric dimension of various coffee compounds

The uniqueness, significance, and applicability of the concentrated study of the edge-based metric dimension of diverse coffee compounds to the fields of graph theory and coffee compounds research, among other things, determines its influence.

Here are some considerations: *Novelty and Contribution:* Researchers in both graph theory and chemistry may be interested in the study's innovative technique or notion about edge-based metric dimension in the context of coffee chemicals. Novelty is frequently a major effect factor. *Interdisciplinary Connection:* Researchers from a variety of disciplines may become interested in the intersection of graph theory and research on coffee chemicals, resulting in collaborations and conversations that close gaps between many scientific groups. *Graph Theory Advancement:* As a result of the study's introduction of novel graph theory methodologies, algorithms, or discoveries pertaining to the edge-based metric dimension, graph theory as a whole may advance. *Insights into Coffee Compounds:* The study shed knowledge on the connectivity patterns of molecules, it might further the study of chemistry or biochemistry by shedding light on the relationships or characteristics of coffee chemicals. *Applications and Practical Relevance:* The study may have potential uses in bioinformatics, drug discovery, or other areas where comprehending molecular connection is crucial.

*Publication and Citations:* The frequency with which a work is cited by other researchers is a common way to gauge its influence. The study might be referenced in subsequent studies if it introduced significant ideas or findings. *Educational Value:* Studies that reveal connections between what at first glance appear to be unrelated subjects can be used as teaching resources, encouraging students to think creatively and recognize the relevance of mathematical ideas across a range of contexts. *Public Engagement:* The study might promote greater public interest in science if it attracted media or public notice as a result of its unusual subject mix.

## 4 Conclusion

In this article, we studied a few coffee chemical structures in terms of edge locating numbers or edge metric dimension. Moreover, these studies contain graphs of the theoretical characteristics of coffee chemical structures. This paper specifically examines the coffee compounds like caffeine, diterpene, or cafestol and kahweol, chlorogenic, caffeic, gallotannins, and ellagitannins. Moreover, the results are concluded in the Table 7.

**Table 7. Summary of the results.**

| $G$ | Metric Dimension | $R_e$ |
|---|---|---|
| $G_{\text{Caffeine}}$ | 2 | $\{v_{10}, v_{12}\}$ |
| $G_{\text{Cafestol}}$ | 3 | $\{v_3, v_{21}, v_{23}\}$ |
| $G_{\text{Chlorogenic}}$ | 3 | $\{v_{18}, v_{19}, v_{21}\}$ |
| $G_{\text{Caffeic}}$ | 2 | $\{v_{11}, v_{13}\}$ |
| $G_{\text{Gallotannins}}$ | 3 | $\{v_{15}, v_{19}, v_{23}\}$ |
| $G_{\text{Ellagitannins}}$ | 3 | $\{v_{27}, v_{29}, v_{33}\}$ |

## Author Contributions

**Conceptualization:** Ali Ahmad.

**Data curation:** Ali Ahmad.

**Formal analysis:** Ali Ahmad.

**Funding acquisition:** Ali N. A. Koam.

**Investigation:** Ali N. A. Koam.

**Methodology:** Ali N. A. Koam.

**Project administration:** Ibtisam Masmali.

**Resources:** Ibtisam Masmali.

**Software:** Ibtisam Masmali.

**Supervision:** Rehab Alharbi.

**Validation:** Rehab Alharbi.

**Visualization:** Rehab Alharbi.

**Writing – original draft:** Muhammad Azeem.

**Writing – review & editing:** Muhammad Azeem.

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
