## [Decision Letter · Decision Letter 0]

17 Aug 2023

PONE-D-23-20879Edge Based Metric Dimension of Various Coffee CompoundsPLOS ONE

Dear Dr. Azeem,

Thank you for submitting your manuscript to PLOS ONE. After careful consideration, we feel that it has merit but does not fully meet PLOS ONE’s publication criteria as it currently stands. Therefore, we invite you to submit a revised version of the manuscript that addresses the points raised during the review process.

Specifically, both reviewers expressed concerns regarding whether the following criteria for publication were met by the submitted manuscript.

Conclusions are presented in an appropriate fashion and are supported by the data.The article is presented in intelligible fashion and is written in standard English.Experiments, statistics, and other analyses are performed to a high technical standard and are described in sufficient detail.Please submit your revised manuscript by Oct 01 2023 11:59PM. If you will need more time than this to complete your revisions, please reply to this message or contact the journal office at plosone@plos.org. Please include the following items when submitting your revised manuscript:A rebuttal letter that responds to each point raised by the academic editor and reviewer(s). You should upload this letter as a separate file labeled 'Response to Reviewers'.A marked-up copy of your manuscript that highlights changes made to the original version. You should upload this as a separate file labeled 'Revised Manuscript with Track Changes'.An unmarked version of your revised paper without tracked changes. You should upload this as a separate file labeled 'Manuscript'.

We look forward to receiving your revised manuscript.

Kind regards,

Eyas Mahmoud

Academic Editor

PLOS ONE

Reviewers' comments:

Reviewer's Responses to Questions

**Comments to the Author**

1. Is the manuscript technically sound, and do the data support the conclusions?

Reviewer #1: Yes

Reviewer #2: No

2. Has the statistical analysis been performed appropriately and rigorously? 

Reviewer #1: Yes

Reviewer #2: N/A

3. Have the authors made all data underlying the findings in their manuscript fully available?

Reviewer #1: Yes

Reviewer #2: Yes

4. Is the manuscript presented in an intelligible fashion and written in standard English?

Reviewer #1: Yes

Reviewer #2: No

5. Review Comments to the Author

Reviewer #1: Manuscript ID: PONE-D-23-20879

Journal: PLOS ONE

Title: Edge Based Metric Dimension of Various Coffee Compounds

Authors: Ali Ahmad, Ali N. A. Koam, Muhammad Azeem, Ibtisam Masmali, and Rehab Alharbi

Coffee includes phenolic acids, diterpenes, and caffeine in addition to being a significant dietary supply of physiologically active substances. Some coffee secondary metabolites may favourably alter a variety of anti-cancer defence mechanisms, claims a particular study. In order to comprehend the mechanics of coffee molecules, this research examined a few coffees chemical structures in terms of edge locating numbers or edge metric size. The graph theoretical features of coffee chemical structures are also included in this study. These papers pay particular attention to the compounds present in coffee, including caffeine, diterpene, kahweol, chlorogenic, caffeic, gallotannins, and ellagitannins. The paper is well written, interested and the results are good, I would like to suggest the following MINOR corrections before acceptance:

ــــــــــــــــــــــــــــــــــــــــــــــــــــــــــــــــــــــــــــــــــــــــــــــــــــــــــــــــــــــــــــــــــــــــــــــــــــــــــــــــــــــــــــــــــــــــــ

(1) A professional proofreading revision is strongly required. Typos must be corrected.

(2) Please add more details about the studied problem

(3) The introduction must be reformulated to contain literature and future works, the main aim of the work. Also, the arrangement of the manuscript should be added in a paragraph at the end of the introduction.

(4) The authors should state clearly in the introduction the advantages of the used technique and a summary of the literature.

(5) The results are interesting, also there was a great effort done, but there are NO clear applications of these results. If the authors can add some applications of their results this will be great

(6) I didn’t see any mathematical model for the problem under study, I think if there is a mathematical model that describes the studied problem it should be added or minimum referring to it in some references.

(7) Some new works on the studied problem should be added, this will improve the paper.

(8) The authors should revise and carefully arrange the references according to the guidelines of the journal.

Thanks a lot, to the editorial board of the PLOS ONE Journal

Reviewer #2: I found the article presents interesting ideas for studding the molecules found within coffee, but I have some major issues with article which prevents it from being published.

The biggest issue is that the article does not motivate the reason why determining the edge dimension of coffee is useful. It is not clear from the article what we gain by knowing the edge dimension of coffee. I would like to see a section which explains the usefulness of edge dimension when it comes to analysing chemical compounds, to at least prime the reader into understanding what these numbers mean when it comes to chemical reactions. In the same vain, there is no discussion of the results in the conclusion which is sorely missed as a discussion of any implications of the results would ground the paper and make it far more impactful.

The article was confusing to read, particularly when it came to explaining the core methodology and core concepts. The explanation of the edge metric of a graph was unclear and by extension the edge dimension as well. Definitions 1.2 and 1.3 where hard to understand and it wasn't really clear what they were defining. The notation used by the definitions was non-standard but did not heavily impact on the clarity of the definitions. Figure 1 describes the methodology but has some quirks/mistakes in it. A step is repeated twice in Figure 1, specifically "Computing the edge resolving the graph," was this step repeated twice? Why was it repeated twice? I am guessing that it is just one step. The authors should make this clearer. In the step "Computing the edge metric dimension using Definition 1.1" Definition 1.1 is used to calculate the edge metric dimension, but Definition 1.1 does not define the edge metric dimension but only defines the edge metric. Presumably the authors meant Definition 1.3.

The article is filled with grammatical errors and typos. Here are a few that stood out to me.

"[21]. fuzzy graph" - "[21]. Fuzzy graph"

"They proved that edge metric dimension" - "They proved that the edge metric dimension"

"the metric di dimension" - "the metric dimension"

I suggest the authors thoroughly proofread the article.

6. PLOS authors have the option to publish the peer review history of their article (what does this mean?). If published, this will include your full peer review and any attached files.

Reviewer #1: No

Reviewer #2: No

---

## [Decision Letter · Decision Letter 1]

13 Nov 2023

Edge Based Metric Dimension of Various Coffee Compounds

PONE-D-23-20879R1

Dear Dr. Azeem,

We’re pleased to inform you that your manuscript has been judged scientifically suitable for publication and will be formally accepted for publication once it meets all outstanding technical requirements.

Kind regards,

Eyas Mahmoud

Academic Editor

PLOS ONE

Additional Editor Comments (optional):

Reviewers' comments:

Reviewer's Responses to Questions

**Comments to the Author**

1. If the authors have adequately addressed your comments raised in a previous round of review and you feel that this manuscript is now acceptable for publication, you may indicate that here to bypass the “Comments to the Author” section, enter your conflict of interest statement in the “Confidential to Editor” section, and submit your "Accept" recommendation.

Reviewer #1: All comments have been addressed

Reviewer #2: All comments have been addressed

2. Is the manuscript technically sound, and do the data support the conclusions?

Reviewer #1: Yes

Reviewer #2: Yes

3. Has the statistical analysis been performed appropriately and rigorously? 

Reviewer #1: Yes

Reviewer #2: N/A

4. Have the authors made all data underlying the findings in their manuscript fully available?

Reviewer #1: Yes

Reviewer #2: Yes

5. Is the manuscript presented in an intelligible fashion and written in standard English?

Reviewer #1: Yes

Reviewer #2: Yes

6. Review Comments to the Author

Reviewer #1: The papaer is modified according to the comments. The authors answered all questions and I think now the paper can be Accepted and published in PLOS ONE journal

Reviewer #2: (No Response)

7. PLOS authors have the option to publish the peer review history of their article (what does this mean?). If published, this will include your full peer review and any attached files.

Reviewer #1: No

Reviewer #2: No

---

## [Editor Report · Acceptance letter]

24 Jan 2024

PONE-D-23-20879R1 

PLOS ONE

Dear Dr. Azeem, 

I'm pleased to inform you that your manuscript has been deemed suitable for publication in PLOS ONE. Congratulations! Your manuscript is now being handed over to our production team.

Kind regards, 

on behalf of

Dr. Eyas Mahmoud 

Academic Editor

PLOS ONE